# Type 2 Diabetes Prediction Model in China: A Five-Year Systematic Review

**DOI:** 10.3390/healthcare13162007

**Published:** 2025-08-15

**Authors:** Juncheng Duan, Norshita Mat Nayan

**Affiliations:** Institute of IR4.0, Universiti Kebangsaan Malaysia, Bangi 43600, Selangor, Malaysia

**Keywords:** China, type 2 diabetes, risk prediction model, machine learning, generalization ability, external validation

## Abstract

**Background:** China has the largest number of patients with type 2 diabetes (T2D) worldwide, and the chronic complications and economic burden associated with T2D are becoming increasingly severe. Developing accurate and widely applicable risk prediction models is of great significance for the early identification of and intervention in high-risk populations. However, current Chinese models still have many shortcomings in terms of methodological design and clinical application. **Objective:** This study conducts a systematic review and narrative synthesis of existing risk prediction models for type 2 diabetes in China, aiming to identify issues with existing models and provide references with which Chinese scholars can develop higher-quality risk prediction models. **Methods:** This study followed the PRISMA guidelines to conduct a systematic search of the literature related to T2D risk prediction models in China published in English journals from October 2019 to October 2024. The databases included PubMed, CNKI and Web of Science. Included studies had to meet criteria such as clear modeling objectives, detailed model development and validation processes, and a focus on non-diabetic populations in China. A total of 20 studies were ultimately selected and comprehensively analyzed based on model type, variable selection, validation methods, and performance metrics. **Results:** The 20 included studies employed various modeling methods, including statistical and machine learning approaches. The AUC values of the models ranged from 0.728 to 0.977, indicating overall good predictive capability. However, only one study conducted external validation, and 45% (9/20) of the studies binned continuous variables, which may have reduced the models’ generalization ability and predictive performance. Additionally, most models did not include key variables such as lifestyle, socioeconomic factors, and cultural background, resulting in limited data representativeness and adaptability. **Conclusions:** Chinese T2DM risk prediction models remain in the developmental stage, with issues such as insufficient validation, inconsistent variable handling, and incomplete coverage of key influencing factors. Future research should focus on strengthening multicenter external validation, standardizing modeling processes, and incorporating multidimensional social and behavioral variables to enhance the clinical utility and cross-population applicability of these models. Registration ID: CRD420251072143.

## 1. Introduction

Diabetes is one of the most serious and prevalent chronic diseases today, with type 2 diabetes (T2D) accounting for more than 90% of all cases [1]. As T2D progresses, patients face a high risk of complications, including blindness, renal failure, myocardial infarction, stroke and premature death, with an 84% higher risk of heart failure compared to non-diabetic patients [2]. According to the International Diabetes Federation’s Diabetes Atlas, 11th edition (2025), an estimated 589 million adults aged 20–79 years had diabetes globally in 2024 (≈11.1% of the adult population, or 1 in 9), and this number is projected to rise to 853 million by 2050. In the same year, diabetes caused 3.4 million deaths globally and caused at least USD 1 trillion in annual healthcare costs. In China, the burden of diabetes is growing even more dramatically, with the number of adults (aged 20–79 years) with diabetes surging from 22.6 million in 2000 to 90 million in 2011, reaching 148 million in 2024, and projected to climb to 168.3 million by 2050. As one of 37 countries in the Western Pacific region, China currently has the largest number of adults with diabetes in the world and the second highest diabetes-related healthcare expenditure in the world [3], a trend that underscores the urgent need to develop precise diabetes prevention strategies tailored to the Chinese population. The clinical symptoms of type 2 diabetes are usually less pronounced. As a result, the disease may be diagnosed several years after its onset, by which time complications have already developed. Thus, delayed diagnosis is a key factor affecting overall disease manageability and risk of complications [4]. The timely screening and management of people at risk for diabetes is important in reducing the incidence of diabetes [5].

The development of effective early prediction models for T2D can help to identify at-risk individuals and provide valuable insights for clinical decision-making, enabling the early implementation of targeted prevention strategies [6]. Globally, the development of diabetes risk prediction models has undergone a gradual evolution from traditional statistical methods to artificial intelligence algorithms. Early models mainly used methods such as logistic regression to construct concise risk scoring tools (e.g., FINDRISC, QDiabetes, and Framingham models), emphasizing interpretability among variables and simplicity of clinical application [7]. With the improvement in computing power and the application of big data technology, a series of machine learning models have emerged in recent years, such as Random Forest, XGBoost, Support Vector Machines (SVMs), and Deep Neural Networks (DNNs), which have demonstrated significant advantages in the modeling of variable interactions, nonlinear relationship capture, and prediction performance [8]. Li [9] highlighted alterations in the TGF-β/Smad, NF-κB, PI3K/AKT and AMPK pathways in diabetic cardiomyopathy, providing a rationale for integrating cardiovascular, molecular and microbial biomarkers into multimodal hybrid models for T2D risk prediction. These personalized prediction tools have shown promising results [10,11], all of which are widely used in epidemiology, clinical screening, and dynamic risk assessment.

However, despite the continuous development of modeling methods, current T2D risk prediction models in China still have major shortcomings in terms of generalization ability and clinical applicability. First, most studies lacked external validation, and models were only tested in the original dataset or similar populations, limiting their ability to generalize to different regions and populations [12]. Second, most studies treat continuous variables (e.g., BMI, blood pressure, etc.) in artificial groups, which facilitates clinical interpretation, but this results in a significant loss of variable information, reduces model prediction accuracy, and is prone to introducing nonlinear relationship bias [13]. In addition, there is a general lack of standardization in the treatment of missing data, often using simple filling or sample deletion, and sensitivity analyses are rarely performed, leading to potential result bias [14]. More notably, social determinants of health (SDOH, e.g., education level, income level, healthcare accessibility, urban–rural differences, etc.) have a significant impact on the risk of developing T2D and management outcomes in Chinese populations. This impact has been grossly underestimated in most of the studies, despite the fact that there is sufficient evidence to show that these factors have important correlations with metabolic chronic diseases [15]. Sung and Lee [16] systematically reviewed the relationship between multiple SDOH and T2D in Asia, emphasizing the need for culturally appropriate interventions and longitudinal studies; Hu [17] constructed and compared five machine learning models based on SDOH data from 26,298 adults in Fujian Province, revealing the relative importance of each SDOH variable in risk prediction; Lan [18] analyzed the impact of SDOH on self-management behaviors in 495 patients with T2D in Zhejiang Province, and found that education, residential environment, and social support were significantly associated with self-care; Zhao [19] conducted an urban–rural difference study in 3225 older adults in Yunnan Province to quantify the role of SDOH factors such as lifestyle and residential environment on T2D and glucose tolerance abnormalities. Meanwhile, more and more studies have begun to explore the predictive value of molecular and microbial markers. Chang [20] systematically evaluated the hypoglycemic effect of Myrica rubra pomace polyphenols (MRPP) in db/db mice, and found that MRPP exerted multiple physiological effects through the PI3K/AMPK signaling pathway and the remodeling of gut flora, providing a new idea of introducing molecular and microbial signatures into a T2D risk model. The problems mentioned above are particularly prominent in China, and the number of relevant studies in China is currently very limited. There is an urgent need to create models suitable for Chinese populations [21].

The main contributions of this study are as follows. First, it systematically reviews and synthesizes research on type 2 diabetes(T2D) risk prediction models developed in China over the past five years, with a focus on model construction methods, key predictive factors, and performance outcomes. Second, it provides a critical evaluation of existing models, highlighting common issues such as data heterogeneity, insufficient external validation, and the inappropriate handling of continuous variables. Third, the study compares traditional statistical approaches with machine learning-based methods, exploring their respective strengths, limitations, and application contexts. Finally, it proposes directions for future research, emphasizing the need to enhance external validation, incorporate social determinants of health variables, and improve data standardization to strengthen both the generalization ability and clinical applicability of the models.

## 2. Methods

### 2.1. Search Strategy

This study was conducted in accordance with the latest PRISMA guidelines 2020 [22]. A PRISMA checklist is provided in Appendix A to ensure comprehensive reporting. This systematic review was pre-designed and registered in the PROSPERO database prior to the commencement of this study (Registration ID: CRD420251072143). To ensure the comprehensiveness and scientific validity of this systematic literature review, a search was conducted in November 2024 in the three major electronic databases, PubMed, Web of Science, and China National Knowledge Infrastructure (CNKI), covering biomedical sciences, computer sciences, and multidisciplinary cross-cutting fields. Through the combined use of these databases, we were able to retrieve a wide range of studies related to type 2 diabetes prediction and early intervention models in China. In terms of search strategy, we designed a comprehensive keyword combination to ensure the accuracy and breadth of the search. The core search terms include ‘type 2 diabetes’ (‘Type 2 Diabetes’ OR ‘T2D’ OR ‘Diabetes Mellitus Type 2’) and terms related to prediction models (e.g., ‘Prediction Model’ OR ‘Predictive Model’ OR ‘Risk Model’). To ensure the geographical and population relevance of the literature, keywords related to China (e.g., ‘China’ OR ‘Chinese’) were also added to the search. Boolean logic operators were used extensively during the search process, e.g., the AND operator was used to connect different topics such as T2D, predictive modeling, and China to ensure that the retrieved literature contained both of these key elements, while the OR operator was used to extend the search to cover different terminological expressions that might appear. See Appendix A for detailed search terms. With this strategy, we were able to find literature related to prediction models and the early intervention of type 2 diabetes in China from multiple dimensions, avoiding missing potentially important studies.

### 2.2. Inclusion/Exclusion Criteria

To ensure that the timeframe of the studies reflects the latest advances in recent years, we set the timeframe of the search from 1 October 2019 to 1 October 2024. This period not only covers research on predictive models and early intervention strategies for type 2 diabetes over the past five years, but also ensures that we have access to the most recent research findings. The choice of this period is based on the observation of the trend of the application of emerging technologies such as machine learning and artificial intelligence in the field of type 2 diabetes, as these technologies have been widely used in prediction modeling and have gradually matured during this period, especially in countries such as China [23], which has a large population base and abundant health data resources. Additionally, the quantity and quality of related studies have increased significantly [24]. In order to screen for literature of high quality and relevance, we set strict inclusion and exclusion criteria:

Inclusion Criteria: (1) The study population was a Chinese population without diabetes mellitus at baseline. (2) The study was directed at constructing predictive models (excluding diagnostic models) for the risk of developing T2D, and the process of model development, validation, and evaluation was described in detail. (3) The type of study design was a cross-sectional, case–control, or cohort study. (4) The study’s purpose was explicitly focused on the development or validation.

Exclusion Criteria: (1) The study was not conducted on the Chinese population. (2) Studies looked at specific high-risk populations (e.g., obese, hypertensive, etc.). (3) Studies had a predictive endpoint of a combined model of multiple diseases that included T2D, but this was not its only outcome (e.g., combined prediction of cardiovascular disease). (4) Studies focused on predictive models of T2D complications (e.g., retinopathy, nephropathy, etc.). (5) Documents were abstracts of international conferences only, without the full text. (6) Studies were conducted at the molecular, cellular, and genetic levels. (7) Publications were duplicates.

### 2.3. Risk of Bias and Applicability Assessment

To ensure the reliability of the review results, we conducted a systematic assessment of the 20 included studies using the PROBAST tool, covering the four major bias domains—participants, predictors, outcome, and analysis. Additionally, three applicability domains—participants, predictors, and outcomes—were assessed for fit. Each domain was labeled with “+” (low risk of bias/high applicability), “−“ (high risk of bias/low applicability), or “?” (unclear), and the overall risk of bias and overall suitability were synthesized based on the ratings of each domain [25]. This process, which is described in more detail in Section 3.4. “Literature quality assessment,” ensures that the quality of the study is fully understood and that the quality of the study can be assessed by considering it at the time of synthesis. This process ensures a comprehensive understanding of the quality of the research and helps to improve the scientific validity and reliability of the conclusions by taking into account potential systematic errors in the synthesis.

### 2.4. Data Synthesis

Two researchers independently screened the literature, extracted information, and cross-checked results; if there was disagreement, a third party was consulted. For the initial screening of the literature, the title and abstract were read first, and after excluding obviously irrelevant literature, the full text was further read through to finalize the inclusion based on the inclusion and exclusion criteria [26]. For literature management, we use literature management software (e.g., EndNote 21.5 or Zotero 7.0.16) to organize and filter the retrieved literature. After completing the literature search and screening, we conducted a systematic data extraction of the included literature. The purpose of data extraction was to collect relevant information from each piece of literature in order to compare and synthesize different studies [27]. Specifically, we designed a standardized data extraction form, including the following core elements: The first element was the basic information in the studies, such as authors, year of publication, and location of the study. This was used in order to clarify the background and spatial–temporal characteristics of each piece of the literature. The second element included the objectives and methodology of the studies, with a focus on extracting specific designs regarding the prediction models of T2D, such as the type of model, selection of predictor variables, data sources, sample size, and validation methods of the model [28]. In addition, the extraction also pays special attention to the demographic characteristics of the study population, such as age, gender, health status, etc., which may affect the applicability and prediction effect of the model [29]. Given the substantial heterogeneity among the included studies in terms of study design, predictor selection, validation methods, and performance metrics, this review only employed narrative synthesis and did not conduct a quantitative meta-analysis.

## 3. Results

### 3.1. Literature Screening Process and Results

The search yielded 1080 relevant studies, which were screened layer by layer, resulting in the inclusion of 20 studies. The literature screening process is shown in Figure 1.

After screening the titles and abstracts, 362 documents proceeded to the full-text assessment phase. According to the pre-defined exclusion criteria in Methods Section 2.2, we reviewed each of these full texts and finally excluded 342 studies. The exclusion categories and corresponding reasons are shown in Appendix A.

Figure 2 shows the five-year trend in published papers. The figure illustrates the popularity of predictive modeling in medical research. Of these 20 articles, the highest number of studies were published in 2024, with 6 (30.0%). This was followed 2020 and 2023, with 4 (20.0%) published. Finally, there was a decreasing number of studies published in 2022, 2021, and 2019, with 3 (15.0%), 2 (10.0%), and 1 (5.0%) published in successive years, respectively. The figure shows that the number of studies on early prediction models for type 2 diabetes first increased over time, gradually decreased after 2020, and then increased year by year, with a general trend of gradual incremental increase, indicating that the topic has become increasingly important to researchers in recent years.

### 3.2. Basic Characteristics of the Included Literature

All 20 included studies were conducted in China; 3 was prospective, and the remaining 17 were retrospective. Sample sizes (excluding missing data) ranged from 936 to 4,075,431 and the number of patients who experienced outcome events ranged from 99 to 301,347.

As shown in Table 1, fasting blood glucose (FBG) ≥ 7.0 mmol/L was used as an observational endpoint in 17 studies, glycosylated hemoglobin (HbA1c) ≥ 6.5% was used in 9 studies, 2 h postprandial glucose (2h-PG) ≥ 11.1 mmol/L was used in 8 studies, and random blood glucose ≥ 11.0 mmol/L was used in 2 studies.

These studies, published between 2019 and 2024, reflect recent advances in T2D risk prediction research in China. The high frequency of FBG ≥ 7.0 mmol/L as an endpoint suggests its leading role in diabetes screening and diagnosis. Several studies have included multiple diagnostic criteria simultaneously, highlighting the multifactorial nature of T2D risk assessment. The wide range of sample sizes and data sources, from national health databases to local hospitals, further enhances the generalizability of the findings.

### 3.3. Basic Features Included in the Prediction Model

#### 3.3.1. Establishment and Validation of the Model

In terms of variable selection methods, among the 20 relevant studies [42] included, the methods of variable selection showed diversified characteristics, which included both traditional statistical methods and some modern machine learning techniques. Overall, 13 studies first conducted univariate analysis to initially screen variables related to diabetes using *t*-tests, chi-square tests, and other means. This link is particularly common in retrospective studies, facilitating the rapid targeting of factors that may have predictive power [50], and multifactor analysis was also applied in 14 studies, mainly based on multivariate logistic regression or Cox proportional risk modeling, to identify independent predictors under the control of other variables and to quantify the relative contribution of each variable to the risk of diabetes incidence. In the further modeling stage, 5 studies used stepwise regression for variable selection, and 6 studies used LASSO regression. For some complex models, Jiang [37] applied SelectKBest, RFE, the Boruta algorithm, and SHAP value interpretation to assess the contribution of variables to the predictive model, taking into account both precision and interpretability. Yang [43], on the other hand, combined meta-analysis and AUC sorting to optimize variables from the evidence base, and these methods were combined with VIF to exclude covariates to construct a representative prediction model.

Overall, 11 studies maintained the continuity of continuous variables, and 9 studies converted all continuous variables into categorical variables. The AUC values of the included models ranged from 0.728 to 0.977, proving that the models in the 20 studies had good predictive performance. In terms of model validation, 0 studies validated the models externally only, 19 studies validated the models internally only, and only 1 study validated the models using a combination of internal and external validation. In addition, 4 studies used the HL goodness-of-fit test to assess calibration. Overall, 14 studies considered model overfitting and calibrated the model accordingly, as shown in Table 2.

Overall, 14 techniques were extracted from the SLR. As can be seen from the results in Figure 3, machine learning (ML) was the preferred choice, with only a few researchers exploring mathematical and deep learning (DL) approaches. Overall, 18 of the studies (90.0%, 18/20) chose ML to construct prediction models for diabetes progression, while 2 articles (10.0%, 2/20) considered mathematics, and 3 papers used DL (15.0%, 3/20). Figure 3 shows the 12 most commonly used models in the field of diabetes progression prediction. The most commonly used method was logistic regression (LR) (75.0%, 15/20). This is followed by Extreme Gradient Boosting (XGB) (30.0%, 6/20) and Random Forest (RF) (30.0%, 6/20), then Decision Trees (DT), Support Vector Machines (SVM), and Light Gradient Boosting Machines (LGBM), which have almost equal distributions (20.0%, 4/20), and then Multilayer Perceptron (MLP) and Cox Proportional Risk (COX), which are also almost equally distributed (10.0%, 2/20). These are followed by Artificial Neural Networks (ANN), K-Nearest Neighbor Algorithm (KNN), Deep Neural Networks (DNN) and Naive Bayes (NB), which are also almost equally distributed (5.0%, 1/20). Table 3 is a fully enriched comparison table used for all twelve modeling methods, with detailed strengths and weaknesses and classic supporting references for each:

#### 3.3.2. Performance of Predictive Factors in the Model and Research Limitations

A compilation and analysis of the predictors from these 20 studies revealed that the predictors included in the models ranged from 8 to 47. The predictors were categorized into three main groups: demographic factors, physical examination indicators, and laboratory tests. Among them, demographic factors such as age, gender, and family history of diabetes were more common, while physical examination indicators such as body mass index (BMI) and waist circumference were more common. Laboratory indicators are common, such as FBG, HbA1c, triglycerides (TG), etc. Most of the studies were based on physical examination or laboratory test data, and widely included traditional biological variables such as age, gender, BMI, fasting blood glucose (FBG), blood lipids (e.g., TG, HDL-C, LDL-C), and indicators of liver and renal function (e.g., ALT, CREA, BUN). These were used as predictors. However, lifestyle variables such as diet, exercise, smoking, alcohol consumption, and sleep, which are closely related to the development of T2D, were only systematically included in a few studies (e.g., Shao et al. and Jiang et al.); key risk factors recommended by the guidelines, such as family history, a history of gestational diabetes mellitus, HbA1c, and OGTT, were omitted from a number of studies or failed to be introduced due to missing data. This limitation in variable coverage makes the predictive power of the model likely to be limited in reality [63]. In addition, about two-thirds of the studies were based on single-city, single-unit, or single-center data, with poorly representative samples and bias problems such as unbalanced sex ratios or incomplete physical examination data, as shown in Table 4.

### 3.4. Literature Quality Assessment

In order to systematically evaluate the methodological quality of the 20 included early T2D risk prediction model studies, we applied the PROBAST tool to assess risk of bias across four domains (participants, predictors, outcome, and analysis) and applicability across three dimensions (participants, predictors, and outcome). The results are summarized in Table 5: only Shao et al. (2020) [36] was rated as low risk of bias (“+”) in all four domains; the remaining 19 studies exhibited high risk of bias (“−“) in at least the analysis domain, yielding an overall high risk of bias classification (“−“). With respect to applicability, all studies received a “+” rating for each dimension, indicating that the selected models broadly align with the characteristics and clinical context of the general Chinese adult population, and thus possess high applicability and strong potential for implementation.

## 4. Discussion

In this paper, we conducted a systematic evaluation of Chinese studies on T2D prediction models. After a staged screening process, 20 studies were finally adopted. Its key findings are shown in Table 6 below. The AUC values of the included models ranged from 0.728 to 0.977, respectively, indicating that the models had good predictive effects. Although the number of studies on T2D risk prediction models in China has continued to grow in recent years, and advanced modeling methods such as machine learning have been gradually introduced; overall, the relevant studies are still in the developmental stage, and have not yet formed a mature system that is widely applicable to clinical practice. All included studies were at high risk of bias due to a variety of factors including optimism bias, the irrational treatment of missing data, the irrational treatment of continuous variables, unstandardized model assessment, and a lack of external validation [64]. This shows that research on T2D prediction modeling in China is still in its developmental stage.

### 4.1. Homogenization of Predictors

As can be seen from Table 3, T2D risk prediction models usually include common predictors such as age, gender, body mass index (BMI), waist circumference, and fasting blood glucose (FBG). On the one hand, this reflects the important early warning value of the above variables in the pathogenesis of T2D, and suggests that clinical staff should pay great attention to the dynamic changes and comprehensive assessment of these indicators in daily screening and management. On the other hand, it also reflects that the current T2D risk prediction models have obvious homogenization in the selection of variables, resulting in the limited differentiation and applicability of the models. Therefore, future studies urgently need to further explore and integrate new personalized risk factors, such as glycated hemoglobin a (HbA1c), a family history of diabetes mellitus, education level, dietary structure, exercise level, sleep quality, mental health status, and socioeconomic factors, on the basis of the traditional predictors in order to improve the predictive performance and relevance of the models, and to explore new personalized predictors that can help to break through existing developmental ‘bottlenecks’. These efforts improve the predictive performance of the model, enhance individualized treatment, and promote the development of T2D risk prediction in the direction of precision and individualization.

### 4.2. Treatment of Continuous Variables and High Risk of Bias

In the development of T2D risk prediction models, the choice between discretizing inherently continuous predictors (e.g., age, BMI, blood pressure) into categories (“binning”) or retaining their continuous form is fundamental, as it directly affects information retention, statistical power, model calibration, and interpretability. Altman and Royston showed that dichotomizing a continuous variable at an arbitrary cut-point can reduce statistical power by up to one-third and introduce artificial threshold effects that impair calibration [65]. By contrast, Harrell recommends preserving continuous predictors and, when needed, applying methods such as restricted cubic splines or fractional polynomials to capture nonlinearity, thereby maximizing information use and improving both discrimination and calibration—provided that sample size and events-per-variable requirements are satisfied (Table 7) [66].

Although binning can be useful when well-established clinical thresholds exist or rapid risk stratification is desired, we advocate that, whenever sample size and events-per-variable criteria allow, researchers should retain continuous predictors and employ semi-parametric approaches to fully leverage data variability, enhance discrimination, and achieve more reliable calibration. Appropriate regularization or resampling (e.g., bootstrap) should accompany such models to guard against overfitting.

On the other hand, there is a high risk of bias in the current T2D prediction studies, most of which only used a single randomization for division into training and test sets without multiple resampling or multicenter external validation, and often did not provide sufficient description of the method of dealing with the missing values, and did not report the calibration curves, Brier scores, or decision curve analyses, which made it difficult to assess the generalization ability and stability of the models. Although most of the studies were highly relevant to the general Chinese adult population in terms of participants, predictors, and outcomes, the practical generalizability of model performance remains limited due to insufficient sample sizes or event-variable ratios, inconsistent variable screening criteria, and short follow-up periods.

### 4.3. Model Validation and Application

The validation process of a model is the main part of assessing performance. A T2D risk prediction model is used to verify whether the model is reliable and generalizable. Of the 20 studies included in this review, the majority of them only performed internal validation, and only one study used a combination of internal and external validation, with most of the internal validation using, e.g., the division of training and test sets, bootstrap resampling, or cross-validation to assess the discriminative and calibrative ability of the model. Among them, AUC (area under the curve) is the most commonly used assessment metric to measure the discriminative ability of predictive models. Regarding external validation, a critical limitation of current Chinese T2D risk prediction models is the general lack of external validation. External validation involves evaluating a model’s performance in an independent dataset distinct from that used for development, which is indispensable for assessing its reproducibility and transportability to new patient populations. This process ensures that key metrics and calibration remain robust [67]. The TRIPOD statement for the transparent reporting of prediction models further identifies external validation as a key requirement to guard against overoptimistic performance estimates and to support clinical applicability [68]. But most models are still limited to data partitioning within the same population or region and lack validation support across populations and regions now. Failure to conduct external validation exposes models to overfitting, whereby spurious associations specific to the development cohort degrade predictive accuracy in other settings [69]. Empirical investigations have demonstrated that model discrimination often declines substantially upon external testing; for instance, Nieboer found that changes in c-statistic values between development and validation cohorts can exceed 0.1, undermining confidence in risk stratification and decision-making [70]. Without rigorous external validation, risk models may generate misleading predictions, erode clinician trust, hinder uptake in clinical guidelines, and ultimately compromise patient care and resource allocation.

In terms of practical application, only a few models have been constructed as column-line graph tools or have visual interfaces to facilitate healthcare professionals to directly assess individual diabetes risk; no studies have reported that the models have been embedded in real clinical pathways or followed up research to observe their long-term intervention effects. Future research should strengthen the following directions after the model has been established: firstly, we must conduct external validation studies in multicenter settings and different populations as most of the current research only stays at the level of internal validation, and external validation studies should be strengthened in the future, especially the assessment of the model generalization ability based on multicenter, large-sample, and heterogeneous population data. In order to ensure the reliability and generalizability of the T2D risk prediction model in different regions and populations in China, it is recommended to perform the following in the multicenter validation design:

1. Select representative centers, urban and rural medical institutions in each of the five major regions, east, west, south, north, and central, to ensure a variety of geographic regions and medical levels.

2. The operation process in each region should be unified. It is necessary to formulate a concise operation manual or document for this. Unified training, unified fasting blood glucose, BMI, blood pressure and other core indicators of the measurement time, method, and instrument can greatly reduce the systematic and human-caused bias.

3. Measure the number of events required by each center according to the principle of “10 new diabetes events correspond to 1 predictor”, and set aside 10% of the lost visit rate.

4. Use a unified electronic data collection platform to upload data, and conduct regular calibration and random checks.

5. Each center independently calculates and summarizes the data from the AUC and calibration curves, and then combines the data and re-evaluates the overall performance of the model to identify regional differences and guide model optimization.

Additionally, it is necessary to develop risk assessment tools that are easy to deploy and use, such as apps, WeChat applets, or scoring modules, integrated into electronic medical record systems; additionally, scholars must to evaluate the intervention value of the model’s predictions in real health management in conjunction with prospective follow-up visits. This will help to meet the needs of medical staff and community diabetes patients for the use of risk prediction tools for the onset of T2D.

### 4.4. Comparison of Traditional Statistical Methods and Machine Learning Prediction Methods

Based on the comparative analysis of the strengths and weaknesses of the models used in the studies included in Table 2, it can be seen that traditional statistical methods and machine learning-based prediction methods have their own features and strengths, which are suitable for different data characteristics and research purposes. Traditional statistical methods, such as logistic regression and the Cox Proportional Hazards Model, have good interpretability and can clearly quantify the independent associations between predictors and disease risk [10]. These methodological models are simple in structure, computationally efficient, and widely accepted in the clinical field, helping researchers and physicians to understand the model inference process [71]. However, traditional statistical methods usually assume linear relationships between variables and have limited modeling power when confronted with high-dimensional data, complex interactions between variables, or nonlinear patterns [72].

In contrast, machine learning-based prediction methods (e.g., Random Forest, XGBoost, Support Vector Machines, Neural Networks, etc.) have powerful pattern recognition and automatic feature extraction capabilities. Machine learning methods are able to capture higher-order interactions and nonlinear features in the data without the need for preset variable relationships, and show better prediction performance in large-sample, multifeature environments [73]. Some algorithms (e.g., LASSO, Elastic Net, Boruta) can also automatically complete variable selection and dimensionality reduction to improve model generalization. However, at the same time, the interpretability of machine learning models is weak, especially in deep learning models, which easily become ‘black boxes’ and face certain obstacles in the promotion and application in clinical practice [74]. In addition, machine learning methods have higher requirements on data quantity, data quality and parameter tuning, and without a strict validation strategy, there is also risk of overfitting and unstable results [75].

Overall, traditional statistical methods are suitable for scenarios with medium sample sizes, a limited number of variables, and studies that focus more on causal inference or clinical interpretability, while machine learning methods are more suitable for tasks with large scale datasets, complex variables, and studies that prioritize prediction accuracy. In the future, the development of T2D risk prediction models can be based on the rational selection of methods based on specific study designs, or combining the advantages of statistical and machine learning methods to develop hybrid modeling strategies with both efficient prediction and interpretability, in order to better serve the needs of clinical screening and individualized management [76].

### 4.5. Contrast with TRIPOD Reporting Standards

TRIPOD (Transparent Reporting of a Multivariable Prediction Model for Individual Prognosis or Diagnosis) is a checklist of 22 items that ensures studies developing or validating risk prediction models fully report critical elements, such as justification of sample size, strategies for handling missing data, discrimination metrics, calibration measures, decision curve analysis, and validation strategies. This allows readers to assess the validity and applicability of the study. In our review, Ouyang [48] did not describe how missing data were handled, in violation of TRIPOD Item 9; Wu [49] only reported discrimination metrics without presenting calibration-in-the-large or calibration slope, as required by Item 14a,b; Ma [45] provided neither justification of the sample size nor number of outcome events, in line with Item 8; and Miao and Zhu [46] only conducted internal validation, with no assessment of independent cohorts, falling short of Item 10. We therefore recommend that future articles on Chinese T2D prediction models rigorously adhere to the full TRIPOD checklist to guarantee comprehensive documentation of study design, analysis methods, and validation procedures. This will enhance reproducibility and facilitate clinical implementation.

### 4.6. Early Intervention Strategies

Early intervention strategies for high-risk individuals with or at risk for type 2 diabetes encompass lifestyle modification, pharmacotherapy, and community-based support designed to interrupt or delay dysglycemia during the critical window before overt disease develops. In the Finnish Diabetes Prevention Study, personalized counseling to reduce caloric and saturated-fat intake while engaging in at least four hours of moderate-intensity aerobic exercise per week yielded a 58% reduction in diabetes incidence over three years among participants with impaired glucose tolerance [77]. The U.S. Diabetes Prevention Program demonstrated that intensive lifestyle intervention achieved a 58% risk reduction over a mean 2.8-year follow-up compared with a 31% reduction with metformin, highlighting the paramount importance of behavioral change in early prevention [78]. Similarly, the Indian Diabetes Prevention Programme (IDPP-1) showed that lifestyle modification alone delayed progression to type 2 diabetes by nearly three years, with adjunctive low-dose metformin further enhancing outcomes [79]. Collectively, these landmark trials underscore that tailored dietary adjustments, regular physical activity, and judicious use of pharmacological support—reinforced through community health education and behavioral coaching—can significantly reduce progression to type 2 diabetes and provide a robust evidence base for implementing tiered early intervention frameworks.

### 4.7. Perspective for Clinical Practice

International evidence underscores the pivotal role of dedicated case managers in T2D care. In the Netherlands, a narrative review of nurses specializing in lifestyle medicine demonstrated marked improvements in glycemic control and cardiometabolic outcomes under nurse-led lifestyle interventions [80]. In Riyadh, a retrospective follow-up study of 3060 patients with poorly controlled T2D who were managed by a multidisciplinary team led by case managers achieved a 15% reduction in mean HbA1c over six months, alongside significant decreases in LDL-C, total cholesterol, and blood pressure [81]. A Saudi Arabian randomized parallel-group trial involving a senior family physician, a clinical pharmacy specialist, a dietitian, a diabetic educator, a health educator, and a social worker reported a 27.1% relative decrease in HbA1c and significant improvements in fasting blood glucose, lipid profiles, and blood pressure over a median 10-month follow-up [82]. Additionally, a Chinese RCT of a nurse-led, integrative medicine–based, structured education program for individuals with newly diagnosed T2D showed a 0.32% reduction in HbA1c and significant improvements in self-management behaviors and self-efficacy at 12 weeks [83].

Translating these international models into the Chinese context will require several coordinated steps. It is essential to establish a formal certification pathway for lifestyle medicine case manager nurses so that they receive standardized training in behavior-change counseling and chronic disease management. Case managers must then be fully integrated into multidisciplinary primary care teams, working alongside physicians, dietitians, and pharmacists with clearly defined roles and workflows. Finally, robust performance dashboards should be implemented to continuously monitor key metrics such as HbA1c levels, complication rates, and healthcare utilization, thereby enabling data-driven quality improvement.

## 5. Implications for Future Research

Building on the issues and recommendations from the preceding discussion, future Chinese type 2 diabetes risk prediction studies should first expand predictor selection to include lifestyle, socioeconomic and behavioral determinants; second, they should employ modeling approaches that balance interpretability with predictive performance; third, they should preserve continuous variables using spline or other smooth methods to avoid information loss; fourth, they should shift validation toward robust multicenter external testing combined with comprehensive calibration and decision curve analyses; and fifth, model outputs should be integrated into user-friendly tools (e.g., column-line graphs or web-based calculators) and combined with early intervention strategies to maximize prevention. Finally, based on the international practice insights from Section 4.7 regarding lifestyle medicine case management nurses and multidisciplinary teams led by case managers, future research should also assess the feasibility, implementation pathways, and actual impact on patient outcomes of these models in China’s primary healthcare settings. By strictly adhering to these principles, future work will improve reproducibility, clinical applicability, and ultimately the accuracy of diabetes risk stratification.

## 6. Limitation

In this systematic evaluation of risk prediction models for type 2 diabetes in China, certain limitations remain. First, this study only included the literature published before October 2024, which may not reflect the latest scientific advances, and as medical research continues to evolve, subsequent findings may influence the current conclusions. Second, due to differences in data processing methods between the original studies, it was difficult for us to perform in-depth analyses at the predictor level, potentially affecting the accurate assessment of the predictive ability of the model; therefore, future studies should adopt standardized methods of data collection and analysis to improve the consistency and comparability of the studies. In addition, this study was primarily based on data from a Chinese population, potentially limiting applicability to other populations and making our findings not necessarily fully representative of individuals from different regions and ethnicities. Despite these limitations, this study provides important insights into the current state of risk prediction models for type 2 diabetes in China. Future studies should expand the literature screening, update the data, optimize the statistical methods, and improve the representation of different populations in order to obtain more comprehensive and reliable results.

## 7. Conclusions

In conclusion, the increasing prevalence of type 2 diabetes in China requires the development of robust and effective predictive models that can be adapted to the unique demographic and cultural characteristics of this population. This systematic assessment revealed significant gaps in current predictive modeling efforts, particularly in the areas of missing external validation, the incomplete coverage of key variables, unstandardized treatment of continuous variables, and weak bias control strategies. Future research must focus on improving modeling methods, enhancing the treatment of continuous variables, and incorporating multidimensional correlated data into predictive frameworks. By addressing these challenges, we can improve the accuracy and applicability of predictive models for T2D, thereby facilitating early detection and intervention strategies. By drawing on international clinical practice experience, we can promote the application of models and case management approaches in local clinical settings. These measures are essential for managing this growing public health crisis. Ultimately, these efforts will help to improve health and reduce the economic burden associated with diabetes.

## Figures and Tables

**Figure 1 healthcare-13-02007-f001:**
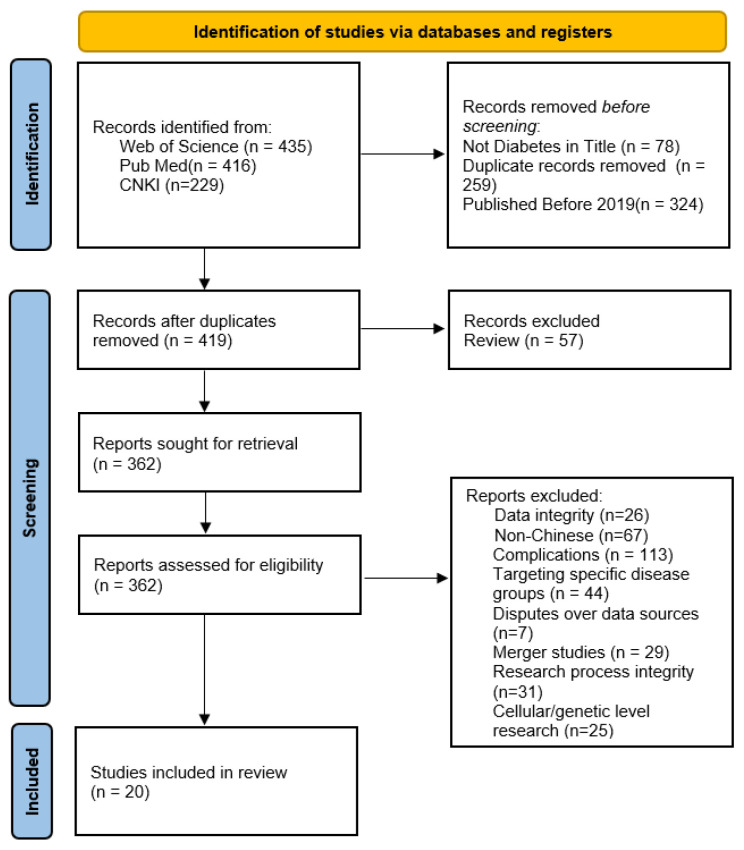
PRISMA flow diagram of article search and selection.

**Figure 2 healthcare-13-02007-f002:**
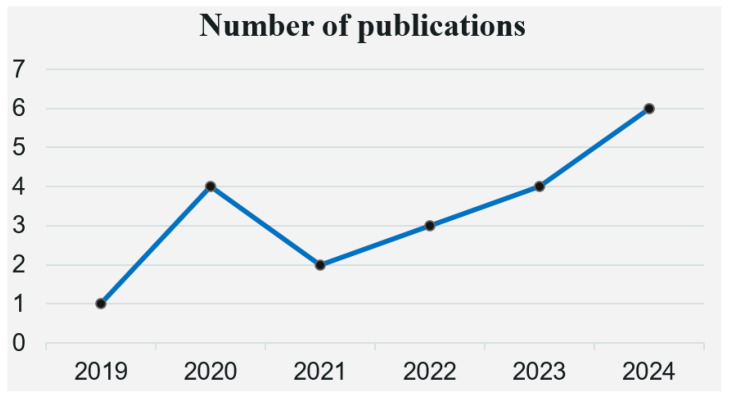
Year-wise distribution of publications relevant to studies.

**Figure 3 healthcare-13-02007-f003:**
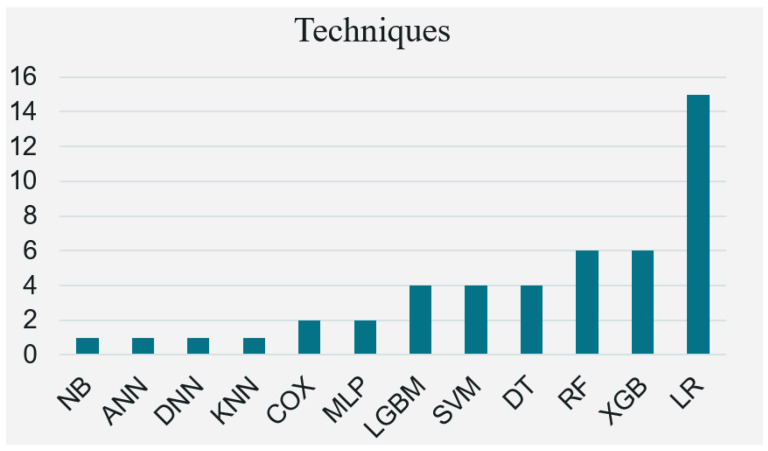
The frequency of prediction models applied by selected articles.

**Table 1 healthcare-13-02007-t001:** Basic characteristics of included studies on risk prediction models for T2D in China.

First Author	Year of Publication (Year)	Country	Research Type	Age of the Research Subjects (Years)	Sample Source	Sample Size	Number of Patients with the Occurrence of Endpoint Events	Observation Endpoint	References
Xu et al.	2022	China	Retrospective study	12~94	Nanjing Drum Tower Hospital Health Management Center	15,166	623	①②⑤	[30]
Lin et al.	2024	China	Retrospective study	≥60	Nanjing Shengrun Hospital	D:1564 V:671	99	②	[31]
Wang et al.	2023	China	Retrospective study	≥18	Monitoring Data on Chronic Disease Risk Factors Among Residents of Dongguan City	4106	149	①③④⑤	[32]
Liu et al.	2024	China	Retrospective study	—	National Health Examination Center Database	D: 32,372 V: 13,875	D: 411 V: 205	①②③⑥	[33]
Yang et al.	2024	China	Retrospective study	≥20	Suzhou University First Affiliated Hospital Health Checkup Center	D: 3221 V: 1381	760	①②③	[34]
Tong et al.	2023	China	Retrospective study	≥18	Sichuan Provincial People’s Hospital	980	513	①②③④⑥	[35]
Shao et al.	2020	China	Retrospective study	20~80	China Health and Nutrition Survey	D: 4498 V: 1525	D: 257 V: 92	①②③④⑤	[36]
Jiang et al.	2024	China	Retrospective study	50~75	Guangzhou Haizhu District Grassroots Community Service Management Information System	252,176	—	①	[37]
Li et al.	2023	China	Retrospective study	≥18	national physical examination (NPE) project	4,075,431	301,347	①③⑤	[38]
Wang et al.	2021	China	Retrospective study	≥18	2018 Health Checkup Data for All Residents of Ili Kazakh Autonomous Prefecture, Xinjiang	D: 366,523 V: 91,630	D: 30,758 V: 7577	①③	[39]
Dong et al.	2022	China	Retrospective study	18~84	Department of Health, Government of the Hong Kong Special Administrative Region	1857	280	①②④⑤	[40]
Liu et al.	2022	China	Retrospective study	≥65	Wuhan Elderly Health Screening Data	127,031	—	①	[41]
Hu et al.	2019	China	Prospective study	—	Retired employees of Dongfeng Motor Corporation (DMC) in Shiyan City, Hubei Province, China	4833	171	①②⑤	[42]
Yang et al.	2024	China	Retrospective study	43~102	Health check-up data for middle-aged and elderly people in Hongguang Street, Pidu District, Chengdu	7602	434	①②④	[43]
Long et al.	2024	China	Prospective study	≥18	Jinchuan Group Staff Hospital	D: 22,025 V: 9438	—	①⑤	[44]
Ma et al.	2020	China	Retrospective study	—	Beijing Huazhao Yisheng Health Checkup Data	D: 4754 V: 2375	—	①	[45]
Miao et al.	2020	China	Retrospective study	—	Physical examination data from a hospital in China	936	—	—	[46]
Ma et al.	2020	China	Prospective study	38~88	A cohort survey of chronic cardiovascular diseases in Fangshan District, Beijing, China	3127	187	①③④⑤	[47]
Ouyang et al.	2021	China	Retrospective study	≥18	Southern Hospital Health Management Center	36,292	2244	①⑤	[48]
Wu et al.	2023	China	Retrospective study	18~80	Fujian Shishi Community Health Center	165,263	—	—	[49]

Note:—indicates no special requirements; D = modeling cohort, V = validation cohort, T2D = type 2 diabetes; ① indicates fasting blood glucose (FBG) ≥ 7.0 mmol/L, ② indicates glycated hemoglobin (HbA1c) ≥ 6.5%, ③ indicates postprandial 2 h blood glucose (2 h-PG) ≥ 11.1 mmol/L, ④ indicates receiving hypoglycemic treatment, ⑤ indicates self-reported diabetes, ⑥ indicates random blood glucose ≥ 11.1 mmol/L; meeting any of the following criteria is sufficient to diagnose T2D.

**Table 2 healthcare-13-02007-t002:** Basic characteristics of development and validation included risk prediction models for T2D in China.

First Author	Modeling Methods	Variable Selection Methods	Methods for Handling Continuous Variables	AUC (95%CL)	Verification Method	Calibration Method
Xu et al.	LR	LASSO regression	Maintain continuity	0.865 (0.847, 0.865)	Internal verification	Calibration curve
Lin et al.	LR	LASSO regression	Maintain continuity	D: 0.824 (0.765, 0.883) V: 0.809 (0.732, 0.886)	Internal verification	Calibration curve
Wang et al.	LR, DT (CART, C4.5), BPNN, SVM, DNN	Univariate analysis, stepwise selection	Maintain continuity	Model 1: 0.962 Model 2: 0.906 Model 3: 0.888 Model 4: 0.977 Model 5: 0.911 Model 6: 0.845	Internal verification	—
Liu et al.	XGBoost, SVM, LR, RF	Univariate and multivariate analysis	Maintain continuity	Model 1: D: 0.986 V: 0.812 Model 2: D: 0.896 V: 0.668 Model 3: D: 0.914 V: 0.913 Model 4: D: 0.998 V: 0.838	Internal verification	Hosmer–Lemeshow test calibration curve
Yang et al.	LR	Univariate and multivariate analysis, stepwise selection	Convert to categorical variable	0.800 (0.770, 0.829)	Internal verification	Calibration curve
Tong et al.	RF, MLP, XGBoost, LGBM, CB	Univariate and multivariate analysis, LASSO regression	Maintain continuity	Model 1: 0.840 Model 2: 0.816 Model 3: 0.848 Model 4: 0.852 Model 5: 0.850	Internal verification	Calibration curve
Shao et al.	LR	Univariate and multivariate analysis, LASSO regression	Maintain continuity	Model 1: 0.788 (0.761, 0.816) Model 2: 0.807 (0.780, 0.834) Model 3: 0.905 (0.879, 0.932) Model 4: 0.882 (0.853, 0.912)	Internal validation and external validation	Calibration curve and bootstrap resampling
Jiang et al.	RF, KNN XGBoost VC	—	Maintain continuity	—	Internal verification	Calibration curve and bootstrap resampling
Li et al.	CART, LGBM, RF, XGBoost TabNet, MLP, LR	Univariate and multivariate analysis	Convert to categorical variable	Model 1: 0.884 Model 2: 0.881 Model 3: 0.873 Model 4: 0.912 Model 5: 0.876 Model 6: 0.875 Model 7: 0.816	Internal verification	Calibration curve
Wang et al.	LR	Univariate, multivariate analysis, LASSO regression	Convert to categorical variable	D: Male: 0.894 Woman: 0.816 V: Male: 0.865 Woman: 0.815	Internal verification	Hosmer–Lemeshow test calibration curve
Dong et al.	LR, XGBoost	Univariate and multivariate analysis, stepwise selection	Convert to categorical variable	Model 1: 0.812 (0.769, 0.853) Model 2: 0.822 (0.779, 0.863)	Internal verification	Hosmer–Lemeshow test Calibration curve
Liu et al.	LR, DT, RF, XGBoost	Univariate analysis, LASSO regression	Convert to categorical variable	Model 1: 0.760 Model 2: 0.728 Model 3: 0.777 Model 4: 0.780	Internal verification	Calibration curve
Hu et al.	Cox	Univariate and multivariate analysis	Convert to categorical variable	D: 0.850 V: 0.830	Internal verification	—
Yang et al.	LR	Univariate and multivariate analysis	Convert to categorical variable	0.794 (0.771, 0.816)	Internal verification	Calibration curve
Long et al.	Cox	Univariate and multivariate analysis	Convert to categorical variable	D: 3 year: 0.783 5 year: 0.825 7 year: 0.842 V: 3 year: 0.782 5 year: 0.805 7 year: 0.807	Internal verification	Calibration curve
Ma et al.	RF, LR, SVM, DT, Naive Bayes (NB)	Multivariate analysis	Maintain continuity	Model 1: 0.931 Model 2: 0.903 Model 3: 0.813 Model 4: 0.776 Model 5: 0.858	Internal verification	—
Miao et al.	SVM (PSO-FWSVM)	Multivariate analysis	Maintain continuity	—	Internal verification	—
Ma et al.	LR	Multivariate analysis and stepwise selection	Convert to categorical variable	Original model: 0,878 (0.853, 0.903) Model 1: 0.880 (0.856, 0.903) Model 2: 0.880 (0.855, 0.903) Model 3: 0.879 (0.854, 0.903)	Internal verification	Hosmer–Lemeshow test Calibration curve
Ouyang et al.	LR, LGBM	Univariate and multivariate analysis, stepwise selection	Maintain continuity	Model1: 0.906 Model2: 0.910	Internal verification	—
Wu et al.	LR LGBM	—	Maintain continuity	—	Internal verification	—

**Table 3 healthcare-13-02007-t003:** Comparative analysis of screened models.

Method	Usage Count	Strengths	Weaknesses	Reference
LR	15	Simple, fast and interpretable Low computational cost Naturally handles binary outcomes	Assumes linear log-odds relationship Cannot capture complex nonlinearity without manual feature engineering Sensitive to outliers and high-leverage points	[30,31,34,51,52]
XGBoost	6	Highly efficient and scalable gradient boosting Ability to be flexible and adjust to mission needs	Demands careful hyperparameter tuning Less transparent than traditional models	[33,35,53]
RF	6	Robust to overfitting Easily handle high-dimensional data and effectively cover complex relationships between variables	Less interpretable than single decision trees Prediction of large datasets is slow and costly	[35,37,54]
DT	4	Easy to visualize and interpret Handles both numeric and categorical inputs	Prone to overfitting without pruning Unstable, small data changes can alter splits	[32,55]
SVM	4	Robust to overfitting; excels in high-dimensional spaces Handles complex, nonlinear patterns via kernel trick	Computationally expensive on large datasets Difficult to interpret model parameters Highly sensitive to kernel choice and hyperparameters	[32,33,56]
LGBM	4	Fast training and low memory footprint Better able to handle large datasets Excellent predictive performance	Can overfit on smaller datasets Less community support and tools than XGBoost	[48,49,57]
MLP	2	Flexible at modeling complex, nonlinear relationships Once trained, enables fast, real-time predictions	Prone to overfitting without sufficient data Demands careful tuning of network architecture and hyperparameters	[38,58]
COX	2	Produces interpretable hazard ratios Widely used in survival analysis for medical research Ability to effectively process incident event data	Assumes proportional hazards, which may not hold Limitations in dealing with complex relationships between predictors	[44,55]
ANN	1	Automatic learning and fitting of complex nonlinear relationships Flexible network structure, can adjust the number of layers and nodes for different problems	Black box model, poor interpretability, not easy to understand the internal decision-making mechanism. Highly sensitive to the amount of data and hyper-parameter settings, easy to overfitting when the sample is insufficient.	[59]
DNN	1	Learns hierarchical features, capturing deep nonlinear pattern Automatically extracts complex interactions without manual feature engineering	Prone to overfitting in complex architectures Computationally expensive and requires large amounts of data ‘Black-box’ nature makes interpretation very difficult	[60]
KNN	1	Simple and non-parametric behavior Can handle multiclass classification problems Robust to outliers and nonlinear relationships	Computationally expensive for large datasets Requires careful selection of the number of neighbors (k) and distance metric Sensitive to irrelevant features and high-dimensional data	[61]
NB	1	Simple and computationally efficient Performs well with high-dimensional data	Assumes proportional hazards, which may not hold Not well-suited for capturing complex relationships	[45,62]

**Table 4 healthcare-13-02007-t004:** Predictors, presentation and limitations of included risk prediction models for T2D in China.

First Author	Predictor Factor	Limitations
Xu et al.	Gender, Age, BMI, ALT, CREA, CHOL, HDL, GLU, MCHC, WBC,	Predicting the risk of type 2 diabetes solely based on laboratory data does not include factors such as diet, exercise, or genetics, which have been proven to be closely related to type 2 diabetes. Single-center data sources, lack of external validation.
Lin et al.	Age, Gender, BMI, FBG, ALT, ALT/AST, BUN, TG, Hb	Single-center data sources, lack of external validation, exclusion of key variables (such as family history and history of gestational diabetes), and the model’s applicability being limited to the elderly population.
Wang et al.	Age, drinking, cereals, potatoes, beans, fruits, eggs, milk, poultry, fish, DBP, FPG, TC, TG, HDL-C, LDL-C	Single-center data sources, lack of external validation. Failure to incorporate common disease risk factors such as genetics and self-care conditions (physical activity, sleep duration, etc.) into the model.
Liu et al.	FPG, Age, TG, ALT, BMI, CR, DBP, gender, family	Lack of external validation, single source of data, non-inclusion of key indicators such as HbA1c.
Yang et al.	Gender, Age, BMI, Blood Glucose, HDL-C, LDL-C, Fatty liver, ALT/AST	Few women were included, resulting in an imbalanced male-to-female ratio. No external validation was conducted. The follow-up period was short.
Tong et al.	FBG, previous HbA1c values, having a rational and reasonable diet, health status scores, type of manufacturers of metformin, interval of measurement, EQ-5D scores, occupational status, Age	The sample size is small, and there are recall biases for some variables.
Shao et al.	Model 1: Age, gender, race, BMI, waist circumference, hypertension Model 2: Model 1 + diet (calories, carbohydrates, protein), exercise, sleep duration Model 3: Model 2 + FPG, HbA1c, TG, LDL, HDL Model 4: FPG, HbA1c, TG, LDL, HDL	The data only comes from the China Health and Nutrition Survey (CHNS), which has issues with re-gional and sample representativeness, and there is a lack of further external validation to assess the model’s general applicability.
Jiang et al.	BMI, age, systolic BP, diastolic BP, staple food, exercise frequency, exercise time	The feature variables are not comprehensive enough.
Li et al.	Gender, age, ethnicity, EH, SS, HTN, CAD, PDM, WC, BMI, WBC, PLT, FBG, ECG, TC, TG, LDL-C, HDL-C	Using cross-sectional data cannot establish causal relationships, and the high heterogeneity and missing rates of health check-up data affect the model’s test effectiveness.
Wang et al.	Age, FHOT, WC, TC, TG, BMI, HDLc, and history of hypertension.	It is not possible to analyze causal relationships from cross-sectional data, the regional limitations of data sources affect generalizability, and the model may not cover all risk factors for type 2 diabetes, which could lead to prediction bias.
Dong et al.	Model 1: Age, BMI, WHR, smoking status, sleep duration, vigorous recreational activity time per week, and fruit consumption per week Model 2: age, BMI, WHR, SBP, waist circumference, sleep duration, smoking status, and vigorous recreational activity time per week	This study did not include key risk factors such as family history of diabetes and gestational diabetes history, and the validation was limited to the same population sample, which restricted the comprehensiveness of the results.
Liu et al.	Age, gender, education, marital status, hypertension, fatty liver, exercise, current smoking, BMI, WC, SBP, DBP, FPG, TC, TG, HDL-C, LDL-C, ALT, AST, TBIL, SCR, BUN, and SUA	Selection bias, omission of certain key risk factors (such as HbA1c and insulin), failure to use OGTT may lead to diagnostic bias, only internal validation was conducted and external validation is lacking.
Hu et al.	Age, gender, BMI, waist circumference, blood pressure, fasting blood glucose, lipid profile (TC, TG, HDL-C, LDL-C), serum uric acid, smoking and drinking status, physical activity, history of hypertension, and family history of diabetes.	Insufficient sample representativeness, lack of important predictive factors, internal validation only, short follow-up period, and potential bias in some self-reported data.
Yang et al.	Age, gender, BMI, waist circumference, triglycerides, HDL-C, smoking status, drinking status, history of hypertension, and family history of diabetes	Insufficient sample representativeness, exclusion of certain key risk factors, limitations of diagnostic methods, internal validation only, and potential biases in lifestyle data.
Long et al.	Sex, age, body mass index, alcohol consumption, alcohol abstinence, hypertension, triglycerides, HDL-C, glutamyl transferase, family history of diabetes mellitus, cholecystitis, gallbladder agenesis.	No external validation; no inclusion of lifestyle variables such as diet and exercise; single source of data.
Ma et al.	Forty-seven characteristics such as blood lipids, urinalysis, liver function, blood pressure, age, gender, and height	Single source of data, lack of external validation, many missing datasets.
Miao et al.	BMI, family history of diabetes, diastolic blood pressure, fasting blood glucose, total cholesterol, triglycerides, LDL, heart rate	Single data source, lack of external validation, risk of bias due to sample imbalance, weak interpretability of features.
Ma et al.	Smoking, history of lipid-lowering drug use, 2h-PG, FPG, BMI, family history of diabetes mellitus, abnormal blood pressure markers, history of hypertension drug use	Lack of external validation and lack of extrapolation; low number of incidence and inaccurate prediction of high risk; continuous variables all categorized for treatment, which may reduce prediction accuracy.
Ouyang et al.	Sex, age, BMI, waist circumference, heart rate, systolic blood pressure, diastolic blood pressure, FBG, uric acid, 4 biochemical indicators, 2 liver function indicators, 2 renal function indicators, and 17 routine blood tests, totaling 34 study indicators, were used as independent variables.	Single source of data, lack of external validation, failure to assess the calibration ability of the model, failure to include variables such as lifestyle behaviors, possible retrospective bias.
Wu et al.	Only 42 characteristics were noted, but no specific	No external validation, no reported AUC, ROC curves, lack of model calibration assessment, lack of model interpretability.

**Table 5 healthcare-13-02007-t005:** Evaluation of risk of bias and applicability of the included literature.

Study	ROB					Applicability			
	Participants	Predictors	Outcome	Analysis	Overall ROB	Participants	Predictors	Outcome	Overall Applicability
Xu et al.	+	+	+	−	−	+	+	+	+
Lin et al.	+	+	+	−	−	+	+	+	+
Wang et al.	+	+	+	−	−	+	+	+	+
Liu H et al.	+	+	+	−	−	+	+	+	+
Yang et al.	+	+	+	−	−	+	+	+	+
Tong et al.	+	+	+	−	−	+	+	+	+
Shao et al.	+	+	+	+	+	+	+	+	+
Jiang et al.	+	?	+	−	−	+	+	+	+
Li et al.	+	+	+	−	−	+	+	+	+
Wang et al.	+	?	+	−	−	+	+	+	+
Dong et al.	+	+	+	−	−	+	+	+	+
Liu et al.	+	+	+	−	−	+	+	+	+
Hu et al.	+	?	+	−	−	+	+	+	+
Yang et al.	+	+	+	−	−	+	+	+	+
Long et al.	+	+	+	−	−	+	+	+	+
Ma et al.	+	+	−	−	−	+	+	+	+
Miao et al.	+	+	?	−	−	+	+	+	+
Ma et al.	+	+	+	−	−	+	+	+	+
Ouyang et al.	+	+	+	−	−	+	+	+	+
Wu et al.	+	+	+	−	−	+	+	+	+

Note ROB indicates risk of bias; − indicates high risk of bias/low applicability; + indicates low risk of bias/high applicability; ? indicates unclear.

**Table 6 healthcare-13-02007-t006:** Summary of findings.

Aspect	Main Findings	Recommendation
Predictor diversity	Predictor variables focused on biological indicators, lack of SDOH (lifestyle factors, socioeconomic factors, etc.)	Add multidimensional predictors
Continuous variable processing	Dichotomizing or categorizing continuous variables leads to information loss and exacerbates bias.	Adopt methods that preserve continuity
Risk of bias assessment	Overall risk of bias is high, affecting model reliability	Strengthen risk of bias control and implement pre-registration and standardized development processes
Model Validation and Application	Validation is mostly focused internally, lacking external multicenter validation and clinically friendly deployment tools	Conducted multicenter external validation and developed easy-to-use interfaces such as line charts, WeChat applets, EHR plug-ins, etc.
Statistics vs. machine learning	Adoption of Machine Learning Methods Growing Rapidly, but Interpretability and Clinical Embeddedness Remain to be Improved	Exploring Interpretable Hybrid Models and Optimizing for Clinical Needs

**Table 7 healthcare-13-02007-t007:** Comparison of binning vs. continuous use of predictors in T2D risk prediction models.

Aspect	Binning	Continuous
Information Retention	Loses within-bin variability	Retains full numeric detail
Statistical Power	Substantially reduces statistical power when categorizing	Preserves full variability, maximizing power
Model Calibration	Risk estimates “jump” at bin boundaries, hindering smooth calibration	Spline- or polynomial-based fits yield smoother, more accurate calibration
Interpretability	Easy to explain cut-points and risk groups	Requires interpretation of coefficients or spline functions
Overfitting Risk	Simpler structure may reduce overfitting	Complex fits need regularization or cross-validation to avoid overfit
Sample Size Needs	Lower requirements but must ensure balanced counts per bin	Requires larger sample and EPV ≥ 10 to support reliable estimation

## Data Availability

Not applicable.

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
