# Peer review of "Type 2 Diabetes Prediction Model in China: A Five-Year Systematic Review"

_healthcare, 2025, doi:10.3390/healthcare13162007_

Round 1

Reviewer 1 Report

Comments and Suggestions for Authors

This manuscript provides a systematic evaluation of type 2 diabetes mellitus (T2DM) risk prediction models developed in China over the last five years. The focus on the Chinese population is timely and important given the high national prevalence of T2DM. The manuscript needs revising as follows.

  1. Provide a table of excluded studies with reasons.
  1. Conduct a risk of bias or quality assessment using a validated tool such as PROBAST (Prediction model Risk Of Bias Assessment Tool).
  2. Present findings in a summary table and incorporate the results into the discussion.
  3. The review mentions variations in model types and variable selection but lacks a clear comparative framework.
  4. Discuss the relative strengths and weaknesses of each modeling approach.
  5. Expand the discussion on the statistical implications of binning vs. continuous variable use.
  6. Cite examples from the included studies and contrast them with guidelines or established modeling standards.
  1. Elaborate on the importance of external validation and the consequences of its absence.
  2. Suggest how future studies might design multi-center validation efforts in China.
  1. Consider referencing the social determinants of health literature relevant to T2DM in China.
  1. The introduction should supplement the current research like, https://doi.org//10.3389/fphar.2024.1416403 , https://doi.org/10.1002/mnfr.202400523, https://doi.org/1097/MD.0000000000040412

Reviewer 2 Report

Comments and Suggestions for Authors

Dear Authors,

the comments in the annex file.

Best

Comments on the Quality of English Language

Native english required

Reviewer 3 Report

Comments and Suggestions for Authors

Type 2 Diabetes Prediction Model in China-A Five-Year Systematic Review

A brief summary

This systematic review of 14 studies on type 2 diabetes mellitus (T2DM) risk prediction models in China highlights promising predictive performance but reveals major shortcomings, including limited external validation, inconsistent variable treatment, and insufficient consideration of lifestyle and socioeconomic factors. The authors call for improved model standardization, broader validation, and the inclusion of multidimensional variables to enhance clinical relevance and generalizability across diverse populations.

General concept comments

Article:

The search method is well-structured and systematic, combining domain-relevant databases and a comprehensive keyword strategy. However, the exclusion of non-English sources and lack of detail on review mechanics may limit completeness and reproducibility. Addressing these issues would further strengthen the validity and scope of the review.

Review:

  • To enhance the depth and relevance of the review, expand database coverage to include Chinese medical databases:
  • Embase (for additional biomedical coverage)
  • CINAHL (for nursing/allied health perspectives)
  • Chinese databases like CNKI or Wanfang for local
  • Add methodological terms related to early intervention strategies
  • No date restrictions are mentioned. Consider justifying the time frame chosen

Specific comments:

  • NA

Round 2

Reviewer 2 Report

Comments and Suggestions for Authors

Dear Authors,

certly in this form the possible international consideration is possible but any conflict rest:

  • Editing: please adopt, as relevants International Guidelinee, T2D and not T2DM in all part of the manuscript (Abstract, Keywords, Text):  https://diabetesjournals.org/care/issue/48/Supplement_1;
  • In the Introduction update epidemiological data;
  • In the methods missing inidcation of the supplymentary (Search strategy and Prisma Check list);
  • Please update the methods section according Prisma Check List;
  • Discussion: in this study missing clinical practice interpretation and international management of care. In this form not suggested possible international consideration. For these reasons, I suggest to devolp a specific section titled "Perspective for Clinical Practice" that could starting extended, with relevant and international references, from the topics "Lifestyle Medicine Case Manager Nurses for Type Two Diabetes", and "Case Manager Led Multi-Disciplinary Team Approach on Glycemic Control for T2D in Primary Care" that extend the interest of data finding in clinical practice and international view;
  • Table 1: I suggest to reported in Supplemenatry Materials;
  • According conclusion follow the previous suggestions;
  • References: according the previous suggestions update the references in clinical and international view. Update the references over 10 years if aren't with high impact of evidence.
  • The corrections according to suggestions poned are fundamental for possible international consideration.

Author Response

Thanks for your suggestion. Please see the attachment.
